# Real-Time Gait Event Detection with Adaptive Frequency Oscillators from a Single Head-Mounted IMU

**DOI:** 10.3390/s23125500

**Published:** 2023-06-11

**Authors:** Matej Tomc, Zlatko Matjačić

**Affiliations:** 1University Rehabilitation Institute Republic of Slovenia Soča, Linhartova 51, 1000 Ljubljana, Slovenia; zlatko.matjacic@ir-rs.si; 2Faculty of Electrical Engineering, University of Ljubljana, Tržaška 25, 1000 Ljubljana, Slovenia

**Keywords:** gait event detection, adaptive oscillators, virtual reality, inertial measurement units

## Abstract

Accurate real-time gait event detection is the basis for the development of new gait rehabilitation techniques, especially when utilizing robotics or virtual reality (VR). The recent emergence of affordable wearable technologies, especially inertial measurement units (IMUs), has brought forth various new methods and algorithms for gait analysis. In this paper, we highlight some advantages of using adaptive frequency oscillators (AFOs) over traditional gait event detection algorithms, implemented a real-time AFO-based algorithm that estimates the gait phase from a single head-mounted IMU, and validated our method on a group of healthy subjects. Gait event detection was accurate at two different walking speeds. The method was reliable for symmetric, but not asymmetric gait patterns. Our method could prove especially useful in VR applications since a head-mounted IMU is already an integral part of commercial VR products.

## 1. Introduction

Gait analysis is an essential tool of modern physical medicine and rehabilitation. To enable comparisons between walking trials and among different individuals, gait is segmented into gait cycles. A gait cycle is defined as a sequence of repetitive events during walking. Conventionally it encompasses everything between two subsequent heel strikes (HSs) of the same leg, marking the 0 and 100% points in the cycle. Despite the undeniable importance of the role of gait analysis in the patient diagnosis and rehabilitation progress assessment, there is, to the best of the authors’ knowledge, no consensus on a measure that would serve as an objective definition of an HS event.

Traditionally, gait analysis was conducted in clinical and laboratory settings using dedicated and expensive equipment, namely force platforms and 3D motion capture systems, consisting of multiple cameras and reflective markers. HSs, used for segmentation of gait into gait cycles for further analysis, were determined using force platforms, where a certain threshold of vertical ground reaction force was set to trigger the event. As the scope of this field of research grew, various other methods using single [1] and split-belt [2] instrumented treadmills and 3D kinematic measurements using motion tracking systems [3,4,5] were also validated. All of the above are now interchangeably used as the gold standard when validating novel methods for gait event detection.

Gait analysis was initially only carried out post hoc on data recorded during walking assessment trials. Soon, researchers and clinicians discovered the potential of incorporating movement-dependent event control in gait training, such as visual and auditory cueing, and the deployment of precisely timed obstacles and perturbations during walking. This led to the development of algorithms for real-time gait event detection. The required equipment was, however, still prohibitively expensive for anything but clinical use, so the scope of possible applications was limited.

Recent advancements in wearable technologies, such as inertial measurement units (IMUs) and shoe insoles with embedded force-sensing resistors, have provided reliable and affordable means for gait analysis. This has opened the floodgates for a wide range of applications that rely on real-time gait event detection, and which are not restricted to clinics: from exoskeletons and active prostheses to training programs for telerehabilitation and even applications outside the field of rehabilitation, such as health monitoring for athletes and the elderly.

Among the novel systems for real-time gait analysis, the overwhelming majority use either IMUs or insole pressure sensors (IPSs) as their sensors of choice [6]. Theoretically IPSs should have higher accuracy but fall short of IMUs in practice due to the lower ease of use, longer setup times, and most importantly, lower durability, since IPSs are constantly subjected to mechanical wear [7]. IMUs are versatile, low-cost, low-energy sensors, which include a gyroscope (measuring angular velocity), an accelerometer (measuring linear acceleration), and a magnetometer (measuring magnetic field). Various setups of IMUs have been proposed to implement gait analysis, ranging from single to multiple IMUs attached to different segments of the human body. Linear acceleration and angular velocity signals have both been utilized for gait analysis, as well as the sensory fusion of all available signals from an IMU [7].

The biggest limiting factors that prevent novel technologies from moving from research laboratories to practical clinical use often revolve around considerations of safety, knowledge required for therapists to use the technology, and plain convenience. An emerging field, helped along by these novel systems for real-time gait analysis, but likely held back by the aforementioned limiting factors, is rehabilitation supported by virtual reality (VR). VR enables the creation of a highly immersive, precisely controlled, and motivating environments. Coupled with real-time gait analysis, VR has the potential to provide a patient with timely and intuitive biofeedback, which is crucial for relearning gait after suffering a neuromuscular impairment. Visual biofeedback has proven to be a successful method for improving various parameters of pathological gait across different patient populations, both within the VR field [8,9] and outside of it [10].

VR-augmented training has slowly been gaining interest among clinicians, but lackluster implementations have often decelerated the spread of the technology [11]; recent studies have indicated that while the use of commercial VR applications has not shown improvements over conventional rehabilitation, the use of custom-built VR applications with specific rehabilitation goals has [12]. Virtual and augmented reality (AR) applications most often use head-mounted displays (HMDs) to create an artificial audio–visual environment. Current implementations of VR and AR applications have mostly focused on upper extremity rehabilitation [12], since tracking is done using handheld controllers, which are typically commercially sold along with HMDs. For gait training, additional devices are used. To take a significant number of steps, patients must walk on a treadmill. For real-time gait analysis, additional sensors (typically IMUs) are attached to the user’s limbs.

Numerous studies have recently proven the effectiveness of IMUs for gait event detection, but the sensors have usually been placed on the lower body [6]. Reference [7] reported that among the various IMU placements, attaching them to the foot or shank is usually preferable, as it results in the least noisy signals. Placing them on the pelvis or the upper body, however, allows for the detection of signal peaks that result from heel strikes from both legs [13]. The authors are aware of only a single research group that has tested the use of a head-mounted IMU for gait analysis. While the signals that occur at HS are attenuated while traveling up the kinematic chain of the human body, they are still detectable, if slightly delayed (∼30 ms) [13]. Regardless of this attenuation and the reflexive compensatory movements of the trunk and neck, which stabilize the head position during gait [14], a frequency component synchronous with the gait cycle should be prominently featured in the vertical and mediolateral accelerations of the head. In the two studies by Hwang et al. [13,15], a combination of a peak-detection and threshold-based algorithm was used for gait event detection. Thresholds were manually adjusted for different gait speeds and extensive filtering was performed on the signal to avoid event skipping or double counting. The head-mounted IMU was primarily seen as a step-counting device and was validated by only using manual step-counting and a pedometer. HS and toe-off events were recorded, but they were only compared to events detected by the IMUs at the feet. The use of a single head-mounted IMU for precise gait event detection has, to the best of the authors’ knowledge, not yet been properly validated.

A single IMU is already embedded in any commercially available VR HMD. We propose using this sensor to perform real-time gait analysis. This would allow for the immediate use of VR technology, often already owned by clinics and individuals, for gait rehabilitation without any additional setup time or therapist training. We acknowledge that the sensor placement is unconventional and will produce signals that are challenging to extract gait events from. That is why, in this paper, we propose a novel algorithm, based on adaptive frequency oscillators (AFOs), which should allow for accurate real-time gait event detection despite the higher noise and delays arising from placing the IMU on the forehead. We hypothesize that the accuracy of the detection will be sufficient for the purposes of VR applications compared to the gold standard methods. To validate the hypothesis, a short validation experiment was conducted. Our goal with the new method is to eliminate the barrier to entry for clinical and at-home use of VR-augmented rehabilitation.

## 2. Materials and Methods

### 2.1. Theory of Adaptive Frequency Oscillators

AFO-based gait-detection algorithms are emerging approaches that have not yet seen widespread use. To help the reader with understanding our algorithm, we provide a brief overview of adaptive oscillators.

AFOs were introduced by Righetti et al. [16,17]. They modified an oscillator perturbed by a periodic driving signal by transforming the frequency parameter ω into an additional system state with its own dynamics. The resulting AFO can synchronize with the driving signal by matching both its frequency and its phase. AFOs were shown to be able to “learn” the features of the driving signals even for pseudo-periodic and noisy signals. By creating a pool of *n* AFOs driven by a common driving signal, we are able to perform the frequency analysis of the driving signal [17], similar to Fourier analysis, in real time. We are also able to generate a real-time continuous estimation of the signal by summing the output signals of AFOs from the pool, akin to a partial sum of *n* harmonics of a Fourier series.

The dynamics of the pool of oscillators are described as, for i=1,… ,n: (1)$$e=u-\hat{u}$$
(2)$$\dot{\alpha_{0}}=k_{0}e$$
(3)$$\dot{\phi_{i}}=i\omega +k_{\phi }e\cos\phi_{i}$$
(4)$$\dot{\alpha_{i}}=k_{\alpha }e\sin\phi_{i}$$
(5)$$\dot{\omega }=k_{\omega }e\cos\phi_{1}$$
(6)$$\hat{u}=\alpha_{0}+\sum \alpha_{i}\sin\phi_{i},$$
where *u* is the driving signal, $\hat{u}$ is the signal estimation (also called the learned signal), and *e* is the error of the estimation. $\alpha_{0}$ is the DC component of the pool of oscillators. $\alpha_{i}$ is the amplitude and $\phi_{i}$ is the phase of the *i*-th AFO. ω is the fundamental frequency of the pool and the frequency at which the first AFO (AFO_1_) oscillates. Coefficients $k_{0},k_{\phi },k_{\alpha }$, and $k_{\omega }$ are used to tune the gains for each state.

### 2.2. AFO-Based Gait Event Detection Algorithm

Signals measured during gait are prime examples of pseudo-periodic noisy signals with clearly dominant harmonic components, whose frequencies match the gait cadences. They can be used as driving signals without the need for any pre-processing, which eliminates any inherent latency. After a few steps, the phase and fundamental frequency of the pool of AFOs should match the phase and frequency of the gait cycle. Gait phase estimation with AFOs is not only in real-time but also continuous. AFOs, while not appropriate for detecting rapid changes in the walking speed or pattern, can be used to reliably analyze any sufficiently periodic gait, healthy or otherwise, especially in cases of patients with generally symmetrically diminished ambulatory functions. Only a handful of research papers have reported using AFOs for gait detection so far, but their results show examples of the successful implementation of this type of algorithm in both normal and abnormal walker populations [18].

We used the mediolateral component of the linear acceleration of the head as the driving signal for the pool of AFOs. The mediolateral component of the linear acceleration was calculated from linear accelerations in the three axes of the IMU and the global orientation of the IMU, which was handled by the proprietary software of the IMU manufacturer. The vertical component of linear acceleration, which was previously used by Hwang et al. [13], was also considered for the driving signal but had shown poorer performance in the preliminary tests.

A pool of n=4 AFOs was created, following Equations (1)–(6). The initial conditions and tuning gains for the oscillators, shown in Table 1, were chosen in such a way that the fundamental frequency of the pool would rapidly converge toward the cadence of gait (within approximately seven strides). After synchronization, a single full gait cycle is completed during one period of the fundamental (i=1) AFO.

Parameter values in Table 1 were chosen after some trial and error. We offer some suggestions for setting up a pool of AFOs that may be useful to other researchers in the field. $\alpha_{0,0}$ should be around the average value of the driving signal. Inspecting the frequency spectrum of a typical signal that we aim to measure gives us information on the number of AFOs that we need in the pool to model the driving signal, as well as what the initial conditions for the amplitude of each AFO ($\alpha_{i,0}$) should be. $\omega_{0}$ should be of the same order of magnitude as the actual cadence of the gait. Tuning gains depend heavily on the amplitude of the driving signal and should be adjusted accordingly when switching the source of the driving signal. $k_{\phi }$ should not be significantly larger than $k_{\omega }$; otherwise, the sinusoidal shape of the AFO outputs is lost.

The phase of a gait cycle is usually expressed as the percentage of time between two consecutive heel strikes. The phase of the AFO has no inherent physical meaning and is adjusted during the process of synchronization with the driving signal. This causes an offset between the two phases, which is dependent upon which physical signal is chosen as the driving signal as well as differences in walking patterns. To eliminate this offset, an additional peak-detection algorithm is run periodically (every 1.5 s) on a 6 s sliding window of the estimated signal. An example of the driving signal overlaid with the estimation within such a window is shown for a representative subject in Figure 1. The peaks in the estimated signal are easily identified since the noise has already been eliminated. This allows for the simplest possible implementation of the peak detection algorithm, where a sample is identified as a local peak (or valley) when it is larger (or smaller) than the two neighboring samples. Peaks and valleys in the driving signal arise at known moments in the gait cycle. In our case, of the mediolateral head acceleration, the last positive peak and negative valley in the estimated signal during each stride correspond to HS events. These matching events can be used to locally determine the offset between the gait phase and the oscillator phase, providing physical meaning to the oscillator phase and allowing for gait event detection in real time. A schematic overview of the algorithm is shown in Figure 2.

### 2.3. Validation Experiment

#### 2.3.1. Subjects

A group of 7 healthy people (2 females and 5 males; age: 36.7 ± 20.8 years; height: 180.9 ± 14.4 cm; weight: 79.3 ± 28.8 kg) volunteered in this study. All participants were informed of the experimental procedures and gave informed written consent prior to the experiment. The experiments were performed at the University Rehabilitation Institute Republic of Slovenia. The Slovenian National Ethics Committee approved the study.

#### 2.3.2. Data Collection

Most VR headset providers do not grant their users with direct access to the raw signals from the embedded IMU. To avoid any preprocessing conducted by the proprietary software of the headset, we attached a separate IMU to the forehead using an elastic headband. We used an MTx-49A53G25 IMU together with the data acquisition device, both made by Xsens. The signals were sampled at 100 Hz. The data were transferred to a PC, where they were processed in real-time using a custom Python script. All contemporaneous data processing was conducted before the arrival of the following sample. The data were also stored for subsequent statistical analysis.

The participants walked on a custom-built instrumented treadmill. The center-of-pressure (CoP) data were sampled at 250 Hz and synchronized with the IMU measurement.

They completed three walking trials—slow, fast, and asymmetric. Each took 3 min to complete. During the slow and asymmetric trials, the treadmill speed was set to 0.6 m/s. During the fast trial, it was set to 1.2 m/s. During the trials, biofeedback was provided to the participants. An image was projected onto a screen in front of the treadmill, showing the CoP trajectory of their last gait cycle. The participants were briefly educated about the characteristic butterfly shape of the CoP trajectory and they used this information to understand the symmetry of their gait. During the slow and fast trials, the participants were instructed to walk normally. Before the asymmetric walking trial, the participants were instructed to adopt an asymmetrical walking pattern by changing the movement of their right leg and continue to use the altered walking pattern for the duration of the trial. They received no instructions regarding cadence, step length, position, or movement of the head.

#### 2.3.3. Data Analysis

CoP data were recorded for subsequent offline gait analysis. An offline peak-detection algorithm on the anteroposterior CoP data was used to detect HSs and as the gold standard for the novel method validation. The timing differences of the 50 consecutive HS events for each leg, as detected by our AFO-based algorithm and the gold standard method, were analyzed. To assure that the steady state was reached both by the walking subject as well as our algorithm, steps for the analysis were taken from the middle of the trials onward. To allow for the comparison across different walking speeds, errors in HS detection $e_{i}$ were computed as percentages of the previous stride durations, using the following equation: (7)$$e_{i}=\left(t_{CoP,i}-t_{AFO,i}\right)/\left(t_{CoP,i}-t_{CoP,i-2}\right).$$
$e_{i}$ is the error in HS detection calculated at the *i*-th step. The numerator $(t_{CoP,i}-t_{AFO,i}$) describes the absolute error, caused by the different timings as detected by the golden standard method $t_{CoP,i}$ and our AFO-based algorithm $t_{AFO,i}$. Negative error values indicate that our algorithm detected the event prematurely. The denominator $\left(t_{CoP,i}-t_{CoP,i-2}\right)$ describes the time of the last stride (note that the stride was completed after two HS events, first with the contra-lateral leg and then with the currently observed leg).

We investigated whether our method performed differently (the error in the HS detection was the dependent variable) based on walking speed (either slow or fast; Factor A), the individual subject’s walking pattern (seven subjects; Factor B), or the interaction between the two using two-way ANOVA.

## 3. Results

Error distributions of HS detection in slow and fast trials are presented as histograms in Figure 3. Since gaits were symmetric, both errors at left and right HSs are shown in the same histogram. The results from each participant share the same color. The HSs were, on average, detected 1.1% (SD = 6.1%) too early in the gait cycle during slow walking trials and 0.6% (SD = 7.0%) too early during fast walking trials. Our AFO-based algorithm has shown similar performance in slow and fast walking (F = 2.21, *p* = 0.14). Results of HS detection differ between individual subjects (F = 15.33, *p* = 0). Interpersonal differences were dependent on the walking speed (F = 19.40, *p* = 0) and appear to be more pronounced in faster walking. Note that if the error was expressed in seconds, it would be half as large in fast walking compared to slow walking.

During asymmetric trials, the peak-detection algorithm for the contemporaneous synchronization of AFO phase with the gait cycle failed. An example of a 6 s sliding window that we attempted to use for peak detection during asymmetric walking is shown in Figure 4. To allow for a simple comparison, the same subject’s signals were shown in this figure and Figure 1.

After the failed attempt of automatically determining the phase offset, raw data recorded during the experiment were used to simulate real-time data collection and analysis, using a modified approach. Instead of the peak-detection algorithm, a manually selected static offset was chosen. The same offset was applied to process all data collected from the participants. The results for left and right HSs are shown separately in Figure 5. The detection of left leg HSs appears to have performed similarly to the symmetric walking trials (mean detection error: 0.77% too late, SD: 5.30%). The detection of right leg HSs, in which participants were instructed to change their movement to create an asymmetric walking pattern, appears to have been mostly unsuccessful.

## 4. Discussion

The purpose of this research was to develop a new algorithm for real-time gait analysis, using a single head-mounted IMU. Our motivation for using this unusual sensor placement was due to the fact that head-worn IMUs are already integrated into widely used commercial VR technology. Such an algorithm could serve as the basis for developing VR applications that focus on gait analysis, which could help with the adoption of VR technology in clinical and at-home rehabilitation.

Our hypothesis is that the accuracy of HS detection will be sufficient for the purposes of VR applications compared to the gold standard methods.

During normal symmetric walking, our AFO-based algorithm detected HS events in real time with good average accuracy compared to an offline CoP-based peak detection (used as the golden standard in this validation experiment), approximately within 1% of the gait cycle, and with moderate precision (standard deviation up to 7%). Whether such errors are acceptable is not well established in the literature. The closest thing to a benchmark in this field [6] is a study by Pappas et al. [19]; the authors implemented a finite state machine to detect gait events with up to 90 ms of delay.

Most studies that deal with gait event detection algorithms report detection errors as the time in milliseconds after the occurrence of the event. The most intuitive and most popular algorithms for gait event detection use empirically determined rules that are based on distinct features of filtered sensor signals that correlate with gait event occurrences (e.g., characteristic peaks in the signal). Despite the popularity of this approach, it suffers from some inherent detection delay; events such as peaks can only be detected after the event has passed, with additional delays originating from the filters when dealing with noisy data. These delays are mostly consistent across walking speeds.

Our algorithm, however, has no such inherent delay. The errors in our algorithm originate mostly from the fact that our assumption of periodicity is violated. Step-to-step changes force the pool of AFOs to constantly adjust the driving signal estimation. Since the IMU is worn on the head, noise originating from the entire kinematic chain also creates a less periodic driving signal. This is a likely explanation as to why we observe similar event detection errors in terms of gait cycle percentage (and not in terms of absolute time) across different walking speeds. Rather than to rule-based algorithms, we can compare our work to an AFO-based algorithm by Yan et al. [20], which reports a maximum RMSE of the phase error at a desired gait event (in our case HS) to be 2.5% while walking on a treadmill. Smaller errors observed in other methods compared to our method are likely due to their use of the hip joint angle and vertical ground reaction force as the driving signals, which contain less noise and fewer harmonic components, making them simpler for the pool of AFOs to learn.

The asymmetric trial was added to the validation experiment in hopes of simulating the asymmetric pathological gait. The walking patterns of the participants were drastically altered and the algorithm was unable to find characteristic peaks of mediolateral accelerations of the head. After choosing a static phase offset, the algorithm was able to detect HSs with comparable accuracy to the symmetric walking on the left (“healthy”) side, but not on the right (simulated “adversely affected”) side. This suggests that an AFO-based algorithm may not be suitable for people with asymmetric walking patterns. Potential difficulties in estimating pathological gaits using AFO-based algorithms were also considered by Yan et al. [20].

### 4.1. Potential Applications

Despite an AFO-based algorithm likely not being effective when dealing with asymmetric pathological gait, it may still be useful in various cases of impairments that do not affect the patient’s gait symmetry, such as Parkinson’s disease, osteoarthritis, Huntington’s disease, diplegic cerebral palsy, aging, and spinal cord injuries. The algorithm may be especially useful for segmenting pathological gait patterns that do not exhibit typical gait events, such as the equinus gait in diplegic cerebral palsy that lacks clear heel strikes.

Our AFO-based algorithm could be used to enrich treadmill training in VR, which has already shown itself to be effective [9], with additional visual or auditory cueing based on real-time gait events. A simplified avatar of the user can be created in the VR and step cadence of the user and the avatar can be matched by synchronizing the walking pattern of the avatar with the user’s estimated gait phase. Whether movement visualization in VR rehabilitation is best conducted using an avatar or otherwise is inconclusive [21]. Since no additional hardware is needed to add an avatar using our method, widespread hypothesis testing is made easier for such research questions.

If a constant treadmill speed is chosen, gait parameters, such as step length, can be calculated from the HS event timings. All relevant spatiotemporal parameters of the gait, except for the step width, can be obtained. Any of those parameters can be targeted by the therapist. In an example application, a therapist could choose an increase in the average step length to be the goal of a training session. Markings could be generated on the virtual floor in front of the avatar, which the patient could reach only by increasing their step length. The relative position of the avatar’s feet and markings would be a clear and intuitive form of biofeedback.

Such training could be especially useful for balance and gait training in the elderly. In this patient population, virtual reality has already shown some promise [22]. A training method that aimed to prevent falls among the elderly is virtual slip training [23]. Virtual slips were induced manually by an operator. The task of triggering slips could be performed by our algorithm.

### 4.2. Limitations

Our algorithm is sensitive to tuning the parameters of AFOs. Different tuning parameters change the speed at which the pool of AFOs learns a new pattern. The quicker the AFOs can learn, the less they are resistant to noise. Determining optimal parameters requires trade-offs and can be an arduous and subjective process.

The AFOs can learn new patterns on the fly, allowing for changes in walking speed. If the transition from one steady-state walking pattern to another is gradual, conducted over multiple strides, the AFOs are able to follow and adapt. Abrupt changes, such as stopping in place, cannot currently be handled by the algorithm; the “inertia” of the AFOs would cause them to produce a continuously diminishing estimation, still triggering HS events seconds after the person has stopped. An additional start–stop mechanism should be implemented for the practical use of this algorithm.

The validation study was conducted on a relatively small group of seven healthy subjects. This is comparable to the average number of participants in other studies in the field [6]. While this may be sufficient as a proof of concept of our approach in healthy populations, no definitive statements can be made about the use of our approach with pathological gait. We recognize that there is a severe lack of validation experiments for novel gait event detection methods in patient populations and that multiple methods have shown themselves to be ineffective when dealing with abnormal gait [24]. By adding the asymmetric walking trial, we were able to garner some insights into how the effectiveness of our approach may change when moving from healthy to patient populations. While the collected information does hold some value, we do not claim that it is an accurate or sufficient substitution for actual validation in patient populations.

While this paper shows that real-time gait event detection using a single head-mounted IMU is possible, we do not foresee our approach being implemented outside the fields of VR and AR. For higher accuracy, and when dealing with asymmetric gait, other sensor placements should be used. Complex protocols that require measurements of body kinematics and kinetics should likewise be conducted with an appropriate multi-sensor setup. The AFO-based algorithm, however, is in no way limited to the input data used in our work, and we encourage other authors to explore the use cases for it with different IMU placements or sensor signals.

## 5. Conclusions

We developed a novel AFO-based algorithm for real-time gait analysis, which can be used to fairly accurately detect gait events using only data measured from a single head-mounted IMU. We validated the novel method on a group of healthy individuals. The method is reliable in detecting gait events during symmetric walking but not during asymmetric walking.

This research expands the usability of commercially available VR technology since an IMU is already a part of any commercially available VR HMD. Our algorithm allows for established rehabilitation methods, such as real-time biofeedback, to be implemented in VR without additional hardware, thereby greatly simplifying the use of VR technology in clinical and, notably, at-home rehabilitation.

## Figures and Tables

**Figure 1 sensors-23-05500-f001:**
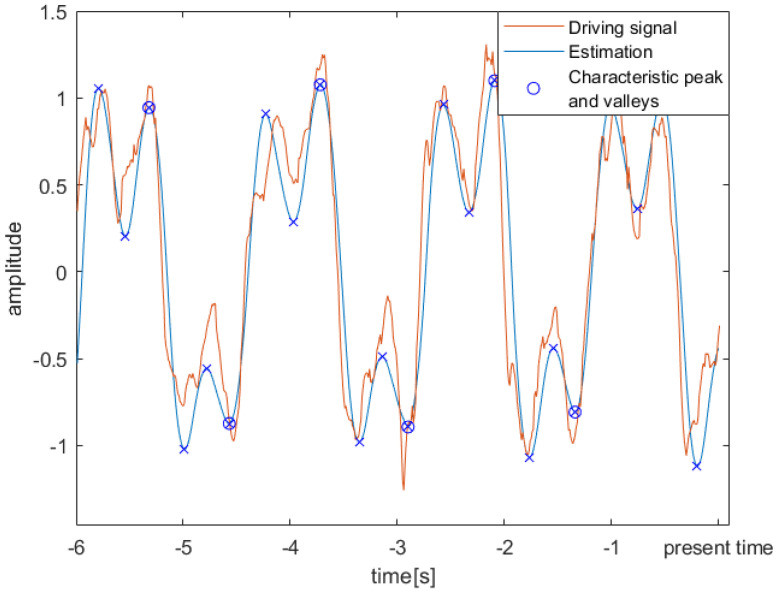
An example of a 6 s sliding window showing the driving signal and the estimation of a representative subject during the slow walking trial. A peak-detection algorithm is run on the estimation to identify the characteristic peaks used to eliminate the offset between the real gait phase and the AFO_1_ phase.

**Figure 2 sensors-23-05500-f002:**
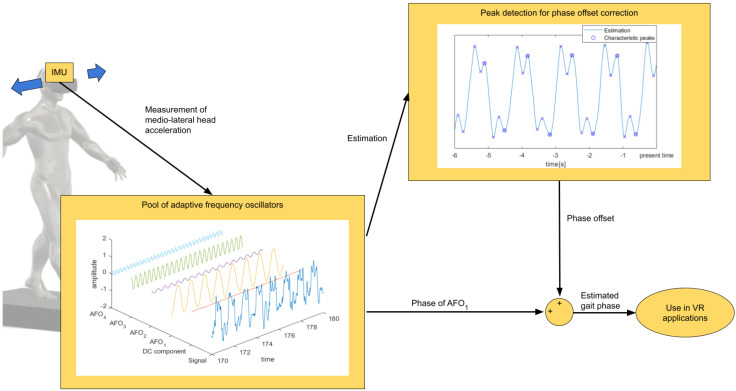
Measurement of the mediolateral head acceleration that is used as a driving signal for the pool of adaptive frequency oscillators (AFO_1_–AFO_4_). A sliding window peak-detection algorithm is then used on the estimation of the signal to find characteristic peaks, correlating to real gait events. Phase offset between the fundamental AFO and actual gait cycle is calculated, allowing for real-time estimation of the gait phase and gait events, which can be used in VR applications.

**Figure 3 sensors-23-05500-f003:**
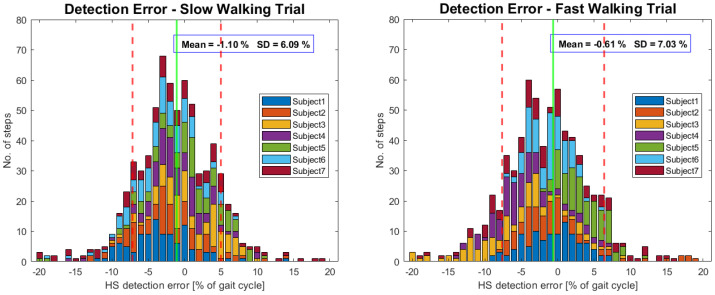
Error distributions of heel strike gait event detection in slow (0.6 m/s) and fast (1.2 m/s) walking trials; 100 steps per participant per trial (50 right and 50 left HSs). Each participant’s steps are shown in the same color in both histograms. Negative values correspond to premature event detection by the AFO-based algorithm.

**Figure 4 sensors-23-05500-f004:**
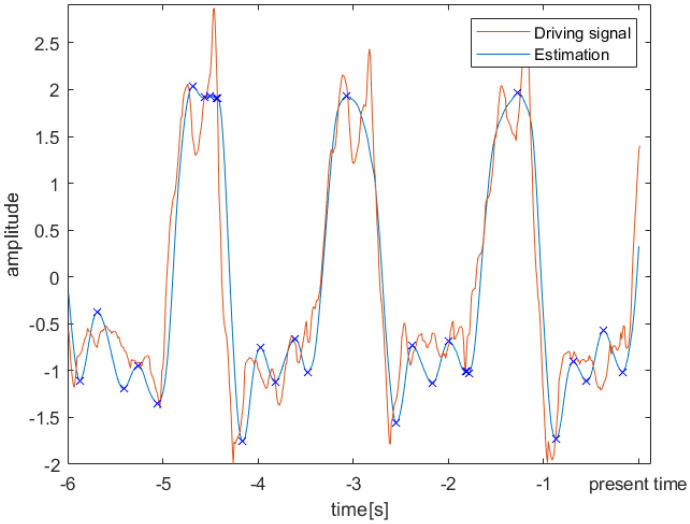
An example of a 6 s sliding window showing the driving signal and the estimation of a representative subject during the asymmetric walking trial. Characteristic peaks that are normally present in the mediolateral head acceleration could not be identified from the estimation.

**Figure 5 sensors-23-05500-f005:**
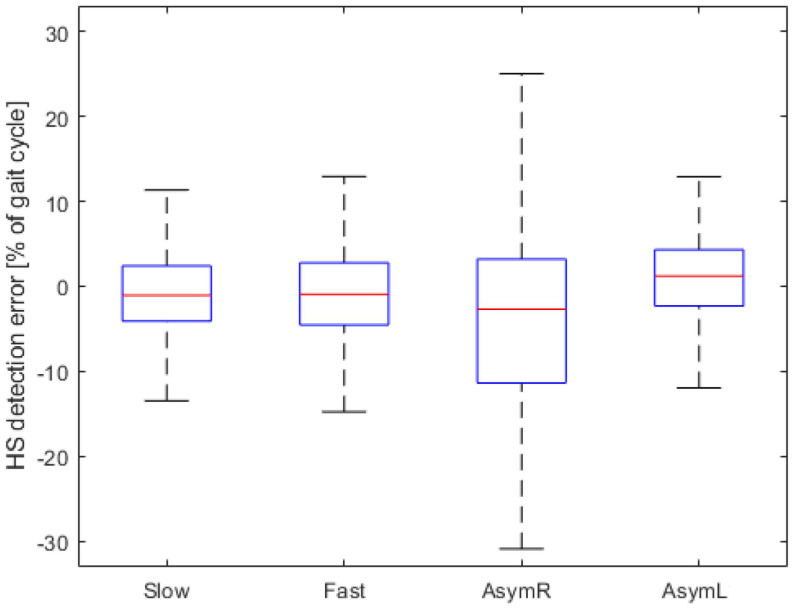
Heel strike detection error distribution across all experimental conditions. Negative values correspond to premature event detection by the AFO-based algorithm. Heel strikes on the left and right sides are shown separately for asymmetric walking trials. Outliers are omitted from the box plot.

**Table 1 sensors-23-05500-t001:** Initial conditions and tuning gains of the pool of adaptive frequency oscillators.

	AFO_1_	AFO_2_	AFO_3_	AFO_4_
** $\alpha_{0,0}$ **	0			
** $\phi_{i,0}$ **	0	0	0	0
** $\alpha_{i,0}$ **	1	0.2	0.5	0.1
** $\omega_{0}$ **	0.9∗2π			
** $k_{0}$ **	0.2			
** $k_{\phi }$ **	3			
** $k_{\alpha }$ **	0.2			
** $k_{\omega }$ **	3			

## Data Availability

The original data will be shared at the author’s discretion. Please contact the corresponding author directly.

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
