# Peer review of "Real-Time Gait Event Detection with Adaptive Frequency Oscillators from a Single Head-Mounted IMU"

_sensors, 2023, doi:10.3390/s23125500_

Round 1

Reviewer 1 Report

1. Compare with existing gait event detection methods: To better understand the algorithm's performance, it would be beneficial to compare it with existing gait event detection methods. This would allow for a more comprehensive evaluation and provide insights into the strengths and limitations of the proposed approach.

2.  Conduct validation experiments with pathological gait: Since the algorithm's performance was limited in simulating pathological gait with asymmetric walking patterns, it would be valuable to conduct validation experiments with individuals who have specific gait impairments or disorders. This would help determine the algorithm's effectiveness in clinical settings.

3.  Explore alternative sensor placements and signals: While the head mounted IMU was used in this study, it would be interesting to explore alternative sensor placements or additional sensor signals for gait analysis. Discussing the advantages and disadvantages of different sensor placements and their potential impact on algorithm performance would enhance the paper's scope.

4. Provide practical applications and implications: Discussing potential practical applications and implications of the algorithm beyond VR and AR fields would be valuable. This could include scenarios such as clinical rehabilitation, monitoring pathological gait, or assisting in the segmentation of atypical gait patterns.

Author Response

Dear reviewer,

we would like to thank you for your careful review and insightful comments. Please find our responses to your points below:

Ad 1.)

Some comparisons to existing gait detection methods are done in the “Discussion” section of the manuscript. There are some radical differences between our approach and most of the popular established methods for gait event detection, so meaningful direct comparison of the result is often not possible. That is why we limited our direct comparisons to article by Yan et al., which is the only one with a method similar enough to ours for fair comparison to be possible.

---

Ad 2.)

We agree that such experimental data would be valuable, however we have ethical concerns with regards to the suggestion. From the perspective of the gait detection algorithm, actual pathological gait would not be treated any differently than asymmetric gait simulated by healthy participants that we measured. As we can already see, pathologies that would create asymmetric gait patterns would likely cause troubles to the algorithm as well. Including patient populations just to confirm this may be problematic, since there is unlikely to be any benefits for the patients themselves or any future patients.

---

Ad 3.)

IMUs are a popular emerging technology and there are numerous studies published with different placements of IMUs all over the body. We recommend further reading of the review article by Prasanth et al. (cited in our work), should you be interested in the topic. A detailed look into different placements is however out of the scope of our article. Still, we agree that the limitations of our approach (in comparison to different placements or multi-sensor setups) could be more clearly presented. To improve this, the last paragraph of the “Limitations” subsection of the discussion was expanded.

Relevant excerpt: 

While this paper shows that real-time gait event detection using a single head-mounted IMU is possible, we do not foresee our approach being implemented outside the fields of VR and AR. For higher accuracy and when dealing with asymmetric gait, other sensor placements should be used. Complex protocols that require measurements of body kinematics and kinetics should likewise be done with an appropriate multi-sensor setup. The AFO-based algorithm however is in no way limited to the input data used in our work and we encourage other authors to explore the use cases for it with different IMU placements or different sensor signals altogether.

---

Ad 4.)

We agree that a few examples of potential use of our algorithm are necessary and have added the subsection “Potential applications” to the discussion. As stated in the “Limitations” subsection however, we do not see much use for this algorithm outside of VR and AR, since the method was developed specifically with the advantages and limitations of VR and AR hardware in mind. For most clinical purposes, other established methods of gait event detection should still be used.

Relevant excerpt: 

Potential applications

Despite an AFO-based algorithm likely not being effective when dealing with asymmetric pathological gait, it may still be useful in various cases of impairments that do not affect the patient’s gait symmetry such as Parkinson’s disease, osteoarthritis, Huntington’s disease, diplegic cerebral palsy, ageing, and spinal cord injuries. The algorithm may be especially useful for segmentation of pathological gait that does not exhibit typical gait events, such as equinus gait in diplegic cerebral palsy that lacks clear heel strike.

Our AFO-based algorithm could be used to enrich treadmill training in VR, which has already shown itself to be effective \cite{yang_virtual_2008}, with additional visual or auditory cueing based on real-time gait events. A simplified avatar of the user can be created in VR and step cadence of the user and their avatar can be matched by synchronizing the walking pattern of the avatar with the user's estimated gait phase. Whether movement visualisation in VR rehabilitation is best done using an avatar or otherwise is inconclussive \cite{ferreira_dos_santos_movement_2016}. Since no additional hardware is needed to add an avatar using our method, widespread hypothesis testing is made easier for such research questions.

If a constant treadmill speed is chosen, gait parameters such as step length can be calculated from the HS event timings. All relevant spatiotemporal parameters of gait, except for step width, can be obtained. Any of those parameters can be targeted by the therapist. In an example application, a therapist could choose an increase in average step length to be the goal of a training session. Markings could be generated on the virtual floor in front of the avatar, which the patient could reach only by increasing their step length. The relative position of the avatar's feet and markings would be a clear and intuitive form of biofeedback.

Such training could be especially useful for balance and gait training in the elderly. In this patient population, virtual reality has already shown some promise \cite{osoba_balance_2019}. A form of training which was performed with the goal of fall prevention of the elderly is virtual slip training \cite{parijat_effects_2015}. Virtual slips were induced manually by an operator. The task of triggering slips could be performed by our algorithm.

Reviewer 2 Report

The submission paper described the real-time detection of gait events from the IMUs, by using the adaptive frequency oscillators (AFOs). The following comments can be considered in the revision manuscript.

1) It is suggested providing the method that how to determine the parameters of AFOs, for example, whether the w0 is associated with the gait speed?

2) The details of peak detection algorithm should be provided in the last paragraph in Page 4.

3) The subjects performed the normal walking during the experiments. How would the abnormal gait affect the peak detection? In Figure 1, it is observed that several peaks in the gait duration should be correctly identified. Whether the altered gait patterns (caused by some diseases) or the altered patterns in the asymmetric walking trials could result in mistakes of gait detection? Please clarify it and add some experiments to demonstrate the results.

4) In the references section, some citations are actually journal papers, not from conference. It is necessary to update the references.

Author Response

Dear reviewer,

we would like to thank you for your careful review and insightful comments. Please find our responses to your points below:

Ad 1.)

A paragraph was added to the “Methods and materials” section (lines 160-169) that provides some explanations as well as some recommendations for choosing appropriate parameters for the AFOs.

Relevant excerpt:

Parameter values in Table \ref{tab1} were chosen after some trial and error. We offer some suggestions for setting up a pool of AFOs that may be useful to other researchers in the field. $\alpha_{0,0}$ should be around the average value of the driving signal. Inspecting the frequency spectrum of a typical signal that we are looking to measure gives us information on the number of AFOs we need in the pool to model the driving signal, as well as what the initial conditions for the amplitude of each AFO ($\alpha_{i,0}$) should be. $\omega_0$ should be of the same order of magnitude as the actual cadence of gait. Tuning gains depend heavily on the amplitude of the driving signal and should be adjusted accordingly when switching the source of the driving signal. $k_\phi$ should not be significantly larger than $k_\omega$, otherwise the sinusoidal shape of the AFO outputs is lost.

---

Ad 2.)

The paragraph in question has been appropriately expanded. Figure 1 was also added to help with the explanation.

Relevant excerpt:

The phase of a gait cycle is usually expressed as percentage of time between two consecutive heel strikes. The phase of the AFO has no inherent physical meaning and is adjusted during the process of synchronization with the driving signal. This causes an offset between the two phases, which is dependent upon which physical signal is chosen as the driving signal as well as differences in walking patterns. To eliminate this offset, an additional peak-detection algorithm is run periodically (every 1.5~seconds) on a 6 second sliding window of the estimated signal. An example of the driving signal overlaid with the estimation within such a window is shown for a representative subject in Fig.\ref{fig1}. The peaks in the estimated signal are easily identified since the noise has already been eliminated. This allows for the simplest possible implementation of the peak detection algorithm, where a sample is identified as a local peak (or valley) when it is larger (or smaller) than the two neighboring samples. Peaks and valleys in the driving signal arise at known moments in the gait cycle. In our case of the medio-lateral head acceleration, the last positive peak and negative valley in the estimated signal during each stride correspond to HS events. These matching events can be used to locally determine the offset between the gait phase and the oscillator phase, providing physical meaning to the oscillator phase and allowing for gait event detection in real time. Schematic overview of the algorithm is shown in Fig.\ref{fig2}.

---

Ad 3.)

From the perspective of the gait detection algorithm, actual pathological gait would not be treated any differently than asymmetric gait simulated by healthy participants that we measured. As we can already see, pathologies that would create asymmetric gait patterns would likely cause troubles to the algorithm as well.

We have expended the “Results” section to more clearly show why the algorithm has trouble dealing with an asymmetric pattern and added additional figure (now Figure 4) to support our claims.

Relevant excerpt:

During asymmetric trials, the peak-detection algorithm for contemporaneous synchronization of AFOs phase with the gait cycle failed. An example of a 6 second sliding window that we attempted to use for peak-detection during asymmetric walking is shown in Fig.\ref{fig4}. To allow for simple comparison, the same subject's signals were shown in this figure and Fig.\ref{fig1}.

(Figure 4 caption: ) An example of a 6 second sliding window showing the driving signal and the estimation for a representative subject during asymmetric walking trial. Characteristic peaks that are normally present in the medio-lateral head acceleration could not be identified from the estimation.

After the failed attempt of automatically determining the phase offset, raw data recorded during the experiment was used to simulate real-time data collection and analysis, using a modified approach. ...

---

Ad 4.)

Thank you for catching this. There seems to be an error occuring with Zotero software, which we used to store our references. We have manually fixed the corrupted entries.

Reviewer 3 Report

The authors have highlighted advantages of using adaptive frequency oscillators (AFOs) over traditional gait event detection algorithms. This work is interesting because AFOs can automatically adjust their frequencies based on the wearer's gait patterns, improving accuracy and adaptability. This potential of AFOs has been explored in this work, particularly in the context of using commercially available VR headsets for rehabilitation applications which can potentially open the door for clinical and at-home use of VR augmented rehab. A few minor comments are as follows — 

  • Per my knowledge, there is an objective definition of heel strike that is widely agreed upon in the field of gait analysis. Heel strike is typically defined as the initial contact of the heel with the ground during the gait cycle. It marks the beginning of the stance phase when the foot makes contact with the ground. Kindly comment as appropriate. 

  • Simplicity, wearability, and real-time analysis are attractive features, however, authors should add more discussion on drawbacks of using a single head-mounted IMU compared to a multi-sensor setup that’ll potentially be more accurate and will be able to better account for asymmetric gait patterns as well. This is probably important for any meaningful real-life use in rehabilitation applications, in my opinion. 

  • Comment on how variability between individuals would be handled. How would any adjusting for variability impact setup and continuous use? 

  • Elaborate on clinical impact of magnitude of errors by providing examples of potential real-life use scenarios. This may lead to coming up with a min. error requirement recommendation for researchers in this field to consider. 

  • Consider adding discussion on how trend analysis can potentially help create a predictive model, so alarming events can be detected in advance and appropriate actionable steps can be implemented to improve clinical outcomes 

  • Elaborate on how sensitivity of head-movements is handled to ensure that motion artifacts don’t interfere with accurate gait detection 

Author Response

Dear reviewer,

we would like to thank you for your careful review and insightful comments. Please find our responses below:

Reviewer's comment: Per my knowledge, there is an objective definition of heel strike that is widely agreed upon in the field of gait analysis. Heel strike is typically defined as the initial contact of the heel with the ground during the gait cycle. It marks the beginning of the stance phase when the foot makes contact with the ground. Kindly comment as appropriate.

Author's response: 

We have chosen awkward wording. The definition of heel strike is indeed as you have stated it, but there is no objective measure to support this definition (is it when the ground reaction force exceeds 10N, 5%, when the heel marker reaches a certain position/velocity etc.; all these definitions are used interchangeably, but they are not quite the same.). We have fixed the wording in lines 19-20 to clarify this.

Relevant excerpt: 

... Despite the undeniable importance of the role of gait analysis in patent diagnosis and rehabilitation progress assessment, there is, to the best of the authors’ knowledge, no consensus on an measure that would serve as an objective definition of a HS event.

---

Reviewer's comment: Simplicity, wearability, and real-time analysis are attractive features, however, authors should add more discussion on drawbacks of using a single head-mounted IMU compared to a multi-sensor setup that’ll potentially be more accurate and will be able to better account for asymmetric gait patterns as well. This is probably important for any meaningful real-life use in rehabilitation applications, in my opinion. 

Author's response:  The last paragraph in “Limitations” subsection was appropriately expanded to cover this.

Relevant exceprt:

While this paper shows that real-time gait event detection using a single head-mounted IMU is possible, we do not foresee our approach being implemented outside the fields of VR and AR. For higher accuracy and when dealing with asymmetric gait, other sensor placements should be used. Complex protocols that require measurements of body kinematics and kinetics should likewise be done with an appropriate multi-sensor setup. The AFO-based algorithm however is in no way limited to the input data used in our work and we encourage other authors to explore the use cases for it with different IMU placements or different sensor signals altogether.

---

Reviewer's comment: Comment on how variability between individuals would be handled. How would any adjusting for variability impact setup and continuous use?

Author's response:  Differences between individuals have shown to have significant effect on the results of our gait event detection algorithm, as we have shown in the “Results” section. As with any repeatable measurement system, a calibration trial could be performed to eliminate bias. Even without the calibration, the errors should not be larger than presented in the paper, so even without any modifications to the algorithm, the error should stay at an acceptable level.

---

Reviewer's comment: Elaborate on clinical impact of magnitude of errors by providing examples of potential real-life use scenarios. This may lead to coming up with a min. error requirement recommendation for researchers in this field to consider.

Author's response: 

A subsection “Potential applications” was added to the discussion, which should illustrate the potential use of our algorithm in real-life scenarios.

We feel that coming up with a minimum error requirement recommendation is outside the scope of our work. We tried our best to find such a recommendation in the review papers of the field, but have found none, so we do not feel comfortable giving such a recommendation ourselves.

Relevant excerpt:

Potential applications

Despite an AFO-based algorithm likely not being effective when dealing with asymmetric pathological gait, it may still be useful in various cases of impairments that do not affect the patient’s gait symmetry such as Parkinson’s disease, osteoarthritis, Huntington’s disease, diplegic cerebral palsy, ageing, and spinal cord injuries. The algorithm may be especially useful for segmentation of pathological gait that does not exhibit typical gait events, such as equinus gait in diplegic cerebral palsy that lacks clear heel strike.

Our AFO-based algorithm could be used to enrich treadmill training in VR, which has already shown itself to be effective \cite{yang_virtual_2008}, with additional visual or auditory cueing based on real-time gait events. A simplified avatar of the user can be created in VR and step cadence of the user and their avatar can be matched by synchronizing the walking pattern of the avatar with the user's estimated gait phase. Whether movement visualisation in VR rehabilitation is best done using an avatar or otherwise is inconclussive \cite{ferreira_dos_santos_movement_2016}. Since no additional hardware is needed to add an avatar using our method, widespread hypothesis testing is made easier for such research questions.

If a constant treadmill speed is chosen, gait parameters such as step length can be calculated from the HS event timings. All relevant spatiotemporal parameters of gait, except for step width, can be obtained. Any of those parameters can be targeted by the therapist. In an example application, a therapist could choose an increase in average step length to be the goal of a training session. Markings could be generated on the virtual floor in front of the avatar, which the patient could reach only by increasing their step length. The relative position of the avatar's feet and markings would be a clear and intuitive form of biofeedback.

Such training could be especially useful for balance and gait training in the elderly. In this patient population, virtual reality has already shown some promise \cite{osoba_balance_2019}. A form of training which was performed with the goal of fall prevention of the elderly is virtual slip training \cite{parijat_effects_2015}. Virtual slips were induced manually by an operator. The task of triggering slips could be performed by our algorithm.

---

Reviewer's comment: Consider adding discussion on how trend analysis can potentially help create a predictive model, so alarming events can be detected in advance and appropriate actionable steps can be implemented to improve clinical outcomes

Author's response: Our algorithm is based on the assumption of periodical movement. Discrete changes that would signify that something irregular is happening and an alarming event might be coming up cannot be evaluated by our method and therefore no predictions can be made.

---

Reviewer's comment: Elaborate on how sensitivity of head-movements is handled to ensure that motion artifacts don’t interfere with accurate gait detection

Author's response: Sensitivity to head-movements is not a major issue, since medio-lateral head acceleration is measured after accounting for changes in orientation of the head. Major acceleration spikes caused by jerky head movements could interfere with the algorithm, but those are not natural movements that we would expect. Even then, this is high frequency noise, that will not be picked up by any of the four AFOs in the pool. In this sense, AFOs work like an implicit Fast Fourier Transform filter, so the method is inherently resistant to noise.

Reviewer 4 Report

I went through the paper and have the following observations:

Please check the title, i.e. why head-mounted is written in small letters.

The authors need to clarify whether in general gait patterns are symmetrical or not in general. Also what are those VR applications where the method can be applied. Also whether these applications still exist and the degree of novelty of this method. 

What is the usefulness of establishing the heel strike while walking?

The results presented in the paper are brief, only 3 figures, and I consider that for a journal paper more results should be presented.

The conclusion part is particularly succinct and inadequate. As I said, the available VR technology should be detailed, and if it is available what is the contribution of the paper? The original contributions made by the paper should be clearly specified.

Author Response

Dear reviewer,

we would like to thank you for your careful review and insightful comments. Please find our responses below:

---

Reviewer's comment: Please check the title, i.e. why head-mounted is written in small letters.

Author's response: This was a typo due to our limited proficiency in the English language and has been fixed.

New title:

Real-time Gait Event Detection With Adaptive Frequency Oscillators From a Single Head-Mounted IMU

---

Reviewer's comment: The authors need to clarify whether in general gait patterns are symmetrical or not in general. Also what are those VR applications where the method can be applied. Also whether these applications still exist and the degree of novelty of this method.

Author's response: 

It is widely accepted that normal gait is symmetrical. Asymmetry is typical in certain pathological gaits. Prominent pathologies that generally result in impaired but still symmetrical gait were previously listed under general discussion, but were now moved under subsection “Potential applications”

We agree that VR applications where the method can be applied should be listed in the article. We have done so in the section “Potential applications”, which also helps illustrate the novel contributions of our method.

Relevant excerpt:

Potential applications

Despite an AFO-based algorithm likely not being effective when dealing with asymmetric pathological gait, it may still be useful in various cases of impairments that do not affect the patient’s gait symmetry such as Parkinson’s disease, osteoarthritis, Huntington’s disease, diplegic cerebral palsy, ageing, and spinal cord injuries. The algorithm may be especially useful for segmentation of pathological gait that does not exhibit typical gait events, such as equinus gait in diplegic cerebral palsy that lacks clear heel strike.

Our AFO-based algorithm could be used to enrich treadmill training in VR, which has already shown itself to be effective \cite{yang_virtual_2008}, with additional visual or auditory cueing based on real-time gait events. A simplified avatar of the user can be created in VR and step cadence of the user and their avatar can be matched by synchronizing the walking pattern of the avatar with the user's estimated gait phase. Whether movement visualisation in VR rehabilitation is best done using an avatar or otherwise is inconclussive \cite{ferreira_dos_santos_movement_2016}. Since no additional hardware is needed to add an avatar using our method, widespread hypothesis testing is made easier for such research questions.

If a constant treadmill speed is chosen, gait parameters such as step length can be calculated from the HS event timings. All relevant spatiotemporal parameters of gait, except for step width, can be obtained. Any of those parameters can be targeted by the therapist. In an example application, a therapist could choose an increase in average step length to be the goal of a training session. Markings could be generated on the virtual floor in front of the avatar, which the patient could reach only by increasing their step length. The relative position of the avatar's feet and markings would be a clear and intuitive form of biofeedback.

Such training could be especially useful for balance and gait training in the elderly. In this patient population, virtual reality has already shown some promise \cite{osoba_balance_2019}. A form of training which was performed with the goal of fall prevention of the elderly is virtual slip training \cite{parijat_effects_2015}. Virtual slips were induced manually by an operator. The task of triggering slips could be performed by our algorithm.

---

Reviewer's comment: What is the usefulness of establishing the heel strike while walking?

Author's response: Heel strike detection is the basis of gait analysis. The heel strike event marks the start of a new gait cycle and is therefore used to segment data into gait cycles for further gait analysis. Wording in the introduction was slightly altered to highlight this.

Relevant excerpt:

HSs, used for segmentation of gait into gait cycles for further analysis, were determined using force platforms, where a certain threshold of vertical ground reaction force was set to trigger the event.

---

Reviewer's comment: The results presented in the paper are brief, only 3 figures, and I consider that for a journal paper more results should be presented.

Author's response: Two additional figures have been added to the paper. The first (now Figure 1) helps with a more detailed explanation of the peak-detection part of the algorithm, which had been somewhat glossed over in the previous version of the manuscript, as was pointed out by another reviewer. We believe that the novel method is the main focus of this paper, more so than the validation experiment, which serves to illustrate the potential usefulness and accuracy of the method. Still, we also added an additional figure (now Figure 4) to illustrate why the algorithm was unsuccessful in asymmetric trials, which is the part of results that was somewhat lacking.

---

Reviewer's comment: The conclusion part is particularly succinct and inadequate. As I said, the available VR technology should be detailed, and if it is available what is the contribution of the paper? The original contributions made by the paper should be clearly specified

Author's response: The paper was expanded with the “Potential applications” section. The excerpt of this section is provided after in our response to your second comment. Conclusion was rewritten to make the original contributions more apparent. In support of both the Potential applications and Conclusions section, part of introduction was also expanded (see below).

Relevant excerpt (updated Introduction and Conclusion sections only):

From Introduction:

... Coupled with real-time gait analysis, VR has the potential to provide a patient with timely and intuitive biofeedback, which is crucial for relearning gait after suffering a neuromuscular impairment. Visual biofeedback has shown itself to be a successful method of improving various parameters of pathological gait across different patient populations both in \cite{booth_immediate_2019,yang_virtual_2008} and outside \cite{franz_real-time_2014} the field of VR.

...

Conclusions

We developed a novel AFO-based algorithm for real-time gait analysis, that can be used to fairly accurately detect gait events using only data measured from a single head-mounted IMU. We validated the novel method on a group healthy individuals. The method is reliable in detecting gait events in symmetric but not asymmetric walking.

This research expands the usability of commercially available VR technology, since an IMU is already a part of any commercially available VR HMD. Our algorithm allows for established rehabilitation methods, such as real-time biofeedback, to be implemented in VR without additional hardware, thereby greatly simplifying the use of VR technology in clinical and especially at-home rehabilitation.

Round 2

Reviewer 2 Report

The authors have improved the manuscript. Regarding the Figure 1 and Figure 4, I would suggest moving the legends outside the figure.